# Align2Concept: Language Guided Interpretable Image Recognition by Visual Prototype and Textual Concept Alignment

## Abstract

Most works of interpretable neural networks strive for learning the semantics concepts merely from single modal information such as images. However, humans usually learn semantic concepts from multiple modalities and the semantics is encoded by the brain from fused multi-modal information. Inspired by cognitive science and vision-language learning, we propose a Prototype-Concept Alignment Network (ProCoNet) for learning visual prototypes under the guidance of textual concepts. In the ProCoNet, we have designed a visual encoder to decompose the input image into regional features of prototypes, while also developing a prompt generation strategy that incorporates in-context learning to prompt large language models to generate textual concepts. To align visual prototypes with textual concepts, we leverage the multimodal space provided by the pre-trained CLIP as a bridge. Specifically, the regional features from the vision space and the cropped regions of prototypes encoded by CLIP reside on different but semantically highly correlated manifolds, i.e. follow a multi-manifold distribution. We transform the multi-manifold distribution alignment problem into optimizing the projection matrix by Cayley transform on the Stiefel manifold. Through the learned projection matrix, visual prototypes can be projected into the multimodal space to align with semantically similar textual concept features encoded by CLIP. We conducted two case studies on the CUB-200-2011 and Oxford Flower dataset. Our experiments show that the ProCoNet provides higher accuracy and better interpretability compared to the single-modality interpretable model. Furthermore, ProCoNet offers a level of interpretability not previously available in other interpretable methods.

## CCS Concepts

• **Computing methodologies** → **Artificial intelligence**; **Object recognition**; **Computer vision problems**; **Image representations**; *Lexical semantics*.

## Keywords

Multimodal Self-explaining Model; Interpretable Image Recognition; Manifold Alignment; Prompt Generation; Large Language Model

## 1 Introduction

Deep learning has demonstrated remarkable performance and has been extensively utilized in various fields, such as image recognition [15, 17] and object detection [11, 36]. However, despite its impressive performance, deep neural networks are still perceived as a black-box model that lacks interpretability. This limitation

*ACM MM, 2024, Melbourne, Australia*

© 2024 Copyright held by the owner/author(s). Publication rights licensed to ACM.
ACM ISBN 978-x-xxxx-xxxx-x/YY/MM
https://doi.org/10.1145/nnnnnnn.nnnnnnn

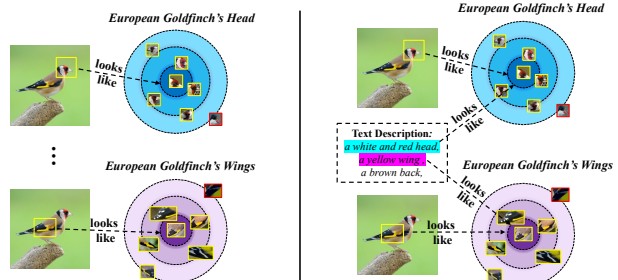

a) Previous prototypical part networks recognize the bird by visual part-level prototypes

b) Our work recognizes the bird by visual prototypes under guidance of conceptual text description

**Figure 1: Different strategies to recognize a bird by semantic concepts. Compared to previous work, our work combines visual prototype with textual concepts to recognize birds.**

restricts their application in high-stakes areas such as finance, self-driving, and disease diagnosis. Therefore, interpretability is crucial in these critical domains, and it is essential to understand precisely how the model makes decisions.

Interpretability of neural networks has recently gained significant attention, and self-explaining convolutional neural networks (CNNs) based on prototype learning [4, 8, 30, 50] have emerged as a major research direction for interpretable computer vision. Rosch [37, 38], the main representative of prototype theory in cognitive science, confirmed through a series of psychological experiments that semantic concepts consist of two factors: the prototype or best instance, and the degree of category membership, which depends on the resemblance to the prototype. Inspired by the prototype theory, the prototypical part network (ProtoPNet [4]) and its extension works [30, 50] first learn the set of prototypes (semantic concepts) from the images in the training set as shown in Figure 1 a) and then make predictions by comparing the similarity between the prototypes and the parts of images. The explanation generated by this method is "the bird is an *European Goldfinch* because parts of the image resemble the prototype parts of *the European Goldfinch*". Visual prototype-based interpretable models can be seen as a type of *introspection* explanation system, which explains how the model determines its final outpute.g., "it is an European Goldfinch because Prototype No.7 has high activation...").

Additionally, there is a *justification* explanation system that generates descriptive sentences detailing how visual evidence is compatible with the system's output (e.g., "it is an European Goldfinch because it has a yellow wing..."). For example, Hendricks et al. proposed [16] focusing on generating sentence-level explanations for object-level images, while Yang et al. proposed [52] focusing on generating phrase-level explanations for object-level images. However, most introspective explanation systems still focus on learning semantic concepts from a single modality (such as images), while

current justification explanation systems lack the part-level visual evidence attribution.

Abundant psychological evidence suggests that humans learn more semantic information by combining language and its meaning in the physical world [12, 33]. In our work, we propose a novel interpretable framework called Prototype-Concept Alignment Network (ProCoNet), which serves as both an introspective explanation system and a justification explanation system. The Prototype-Concept Alignment Network aligns visual part-level prototypes with textual concepts as illustrated in Figure 1 b). Through this network, we can not only determine that this bird is identified because a prototype has high activation but also explain what human-interpretable textual concept corresponds to this highly activated region. Specifically, ProCoNet consists of three spaces: a vision space for extracting prototype region representations, a human-interpretable textual concept space, and a multimodal space for aligning visual prototypes with textual concepts. Firstly, to generate textual concepts that can match visual prototypes, our approach leverages in-context learning and prompt generation strategies with large language models (LLMs). By using pre-annotated textual concepts as contextual clues, our method enables the LLM to directly produce relevant and concise concept descriptions. Then, we employ a prototypical part network [4] to extract regional representations of prototypes in the vision space.

A major challenge is that aligning fine-grained visual prototypes with textual descriptions of concepts is difficult without explicit supervisory signals. To address this issue, we utilize the multimodal space of the pre-trained vision and language model CLIP [34] as an intermediary bridge, employing manifold alignment to learn mapping from the vision space to the multimodal space. Figure 2 illustrates the basic idea of manifold alignment, which aims to construct relationships between different datasets and project data points to a common subspace while preserving local geometry structures. Specifically, the regional features of the prototypes and the features of the region images encoded by the CLIP image encoder reside on different but semantically highly correlated manifolds, i.e., a multi-manifold distribution. To align the both features, we adopt the manifold alignment method by learning a projection matrix that project them into a common subspace. In this subspace, we enforce both features that share similar semantic meanings to be close, i.e. feature matching, while preserving their neighborhood relationships in the original feature spaces, i.e. local geometry preserving. We convert the manifold alignment problem into an optimization problem with orthogonality constraints on the projection matrix, taking into account the criteria of feature matching and geometry preserving. We then transform the optimization problem with Cayley transform and update the the projection matrix using a curvilinear search on the Stiefel manifold. Through the feature transformation of the projection matrix, visual prototypes and text concepts encoded by CLIP text can be aligned within the same space.

We have conducted extensive experiments on two case studies, bird species identification and flower species identification, to evaluate the performance of our model in terms of both interpretability and accuracy. To measure interpretability, we compared the output region of semantic concepts with annotated object parts, which provides a quantifiable metric that was previously missing from

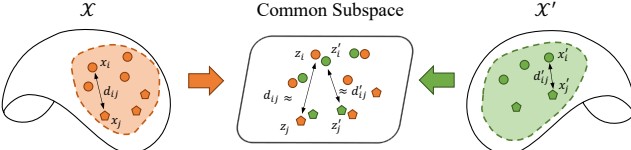

**Figure 2: The basic idea of Manifold alignment: corresponding data points are mapped from two different manifolds to similar locations in a common subspace while preserving the local geometry of each manifold.**

ProtoPNet-based methods. Furthermore, we designed a baseline model that learns concepts directly from features without manifold alignment to demonstrate the effectiveness of our proposed framework. Our model outperformed classical interpretable models in both accuracy and interpretability.

The contributions of our work are summarized as follows:

- A new framework of interpretable image recognition guided by language is first proposed that can learn semantics concepts from both visual and linguistic modality. To my knowledge, this is the first interpretable framework that is both an introspective explainable system and a justification explainable system.
- To obtain textual concepts related to visual prototypes, we developed an innovative prompt generation strategy that integrates in-context learning, prompting large language models to directly produce text concepts that are visually recognizable without the need for post-processing. This significantly simplifies the concept extraction process and enhances the relevance and clarity of the generated descriptions.
- We design a manifold alignment method to optimize the projection matrices on the Stiefel manifold by the Cayley transform, which transforms the features of multi-manifold distributions into a common subspace. Besides, an alternating optimization algorithm is proposed to optimize objectives of manifold alignment and concept learning.
- We apply quantitative metrics of interpretability for the ProtoPNet-based methods. Extensive experiments have demonstrated that reasonable utilization of natural language guidance can improve the accuracy and interpretability of the self-explaining model.

## 2 Related Work

Our work is related to interpretability in image recognition and manifold alignment. In this section, we will give a brief overview of related works.

**Interpretability in Image Recognition.** Previous methods of interpreting deep neural networks can be broadly classified into two types, one devoted to visualizing the underlying patterns of the black-box model, and the other focusing on modeling clear semantic representations. There are a lot of visualization methods to compute the relevance score at each pixel-level location of the image, including the saliency-based methods [7, 26, 54, 57], perturbation-based forward propagation methods [9, 10], and backpropagation-based

methods [2, 28, 42–45, 55]. But the collection of pixels cannot build the connection with semantics concepts of humans and lack of illustrating knowledge hidden behind all activations. Therefore, some methods aim to learn semantics concepts to construct a self-explaining model during the training process, such as [5, 24, 56] focus on learning interpretable filters by restricting each filter response to a specific concept. For grasping the concepts, some researchers try to represent the semantics patterns as the prototype vectors [4, 8, 22, 29, 30, 50]. [4] proposed a prototypical part network (ProtoPNet) to learn the prototypes and can generate a part-level attention map as an explanation by calculating the distance between an input image and learned prototypes. Further, the ProtoPNet has been extended many times [8, 22, 29, 50] and different methods have added different properties to the prototypes. [50] designed a plug-in transparent embedding space, which is spanned by disentangled basis concepts on the Grassmann manifold. Nauta et al. proposed a Neural Prototype Tree that combines prototype learning with decision trees and can explain a prediction by outlining a decision path. However, the above methods only learn semantic concepts from visual modality and lack other modal information to guide concept learning. In contrast, our model is the first framework to learn the visual prototypes guided by textural concepts and a manifold alignment method is proposed to align the multi-modal manifold distribution. In this way, the concepts can capture richer semantic information and provide better interpretability.

Another class of explainable methods involves constructing Concept Bottleneck Models (CBMs) [39] to establish relationships between images and human-understandable attributes. Specifically, these models introduce a bottleneck layer composed of human-understandable attributes between the input and prediction, and predictions are then made by linearly combining these attributes. However, CBMs often require costly attribute annotations. To address this issue, Post-hoc Concept Bottleneck Models (PCBMs) [53] have been proposed, which utilize information from static knowledge bases. Additionally, LaBo [52] has suggested using GPT-3 [3] to first extract sentences containing attribute information about categories, followed by manually annotating some sample pairs to fine-tune a T5 model [35] to segment sentences into smaller attribute descriptions. Ultimately, an optimizable submodule is trained to filter out the most relevant concept sets. However, the recent paper [27] have conducted a thorough analysis of CBMs using post hoc interpretability methods, revealing that CBMs may not effectively correspond to semantically meaningful representations in input space, thereby questioning their current utility and their ability to achieve their intended goals of interpretability, predictability, and intervenability. In our work, we have designed a prompt generation strategy that allows GPT to produce textual concepts focused on prototype descriptions. This approach reduces the dependency on costly attribute information and eliminates the need for complex post-processing. Additionally, because ProCoNet possesses introspective capabilities, it avoids the traditional bottleneck layer issue of failing to generate expected visual explanations.

**Manifold Alignment.** Manifold alignment has been widely used in many fields of machine learning and data mining. The main objective of manifold alignment is to align the sets of data from different manifold distributions by matching the corresponding instances of different manifold distributions and preserving the local geometry of each manifold. Depending on whether the correspondence information is available or not, manifold alignment can be classified into semi-supervised manifold alignment [13, 25, 49] and unsupervised manifold alignment [6, 20, 32, 48]. Generally, the manifold alignment problem is transformed into an optimization problem that finds the projection matrices from the original spaces to a common subspace, and the projection matrices are solved in closed form by eigenvalue decomposition like Canonical Correlation Analysis (CCA) [14]. Different from existing methods, we develop the optimization algorithm of the projection matrices based on the Cayley transform for preserving the orthogonality constraints, which can obtain the global optimal solution [51]. Our model doesn't require the region images of prototypes for guidance during inference, whereas the traditional CCA method requires guidance from corresponding features in multimodal space during testing.

## 3 Method

In this section, we will introduce our proposed framework named Prototyp-Concept Alignment Network (ProCoNet) for image classification in Section 3.1, and the overall architecture is shown in Figure 3. For constructing textual concept space, we crafted a strategy that employs in-context learning techniques for generating prompts, which are then fed into the large language model GPT-3 [3]. This approach generates a diverse set of textual concepts for each category, which are amenable to visual recognition (Section 3.2), and then we employ a pretrained vision-language model, CLIP [34] to encode these textual concepts to multimodal space. Additionally, we propose a manifold alignment method that utilizes the CLIP visual embeddings of cropped regions consistent with visual prototypes to project visual prototype representations into a multimodal space, while preserving the local geometry of the vision space (Section 3.3). In the multimodal space, the projected visual prototype embeddings and textual concept embeddings can compute their mutual similarity and derive scores for each concept. We combine the scores of different concepts through learning a class-concept weight matrix, applying linear weighting, to ultimately obtain the predicted logits in the final classification. Finally, Section 3.4 introduces the implementation details of the two-step alternating optimization process in ProCoNet.

### 3.1 ProCoNet Framework

Figure 3 illustrates the framework of our ProCoNet. Given the image set $\{I_v\}_1^N$ with labels $\{y\}_1^N$ and the textual concept $\{I_l\}_1^{N_b}$, the objective of the ProCoNet is to learn part-level visual prototype $\mathbf{B}$, the projection matrix $\mathbf{P}$ which projects visual features into the multimodal space, and a decision function $\hat{y} = \psi(I_v, I_l; \mathbf{B}, \mathbf{P}, \theta)$, where $\theta$ is the paramerters of $\psi$. Specifically, we can decompose the function into four parts, region feature extraction of visual prototypes, region feature projection by manifold alignment, textural conceptual space construction, and visual prototypes and textual concepts alignment.

**Region Feature Extraction of Visual Prototypes.** In the vision space, we first pre-trained a standard ProtoPNet [4] on the image dataset to learn visual part-level prototypes. Specifically, we denote the input image as $I_v$ and its features extracted by the vision

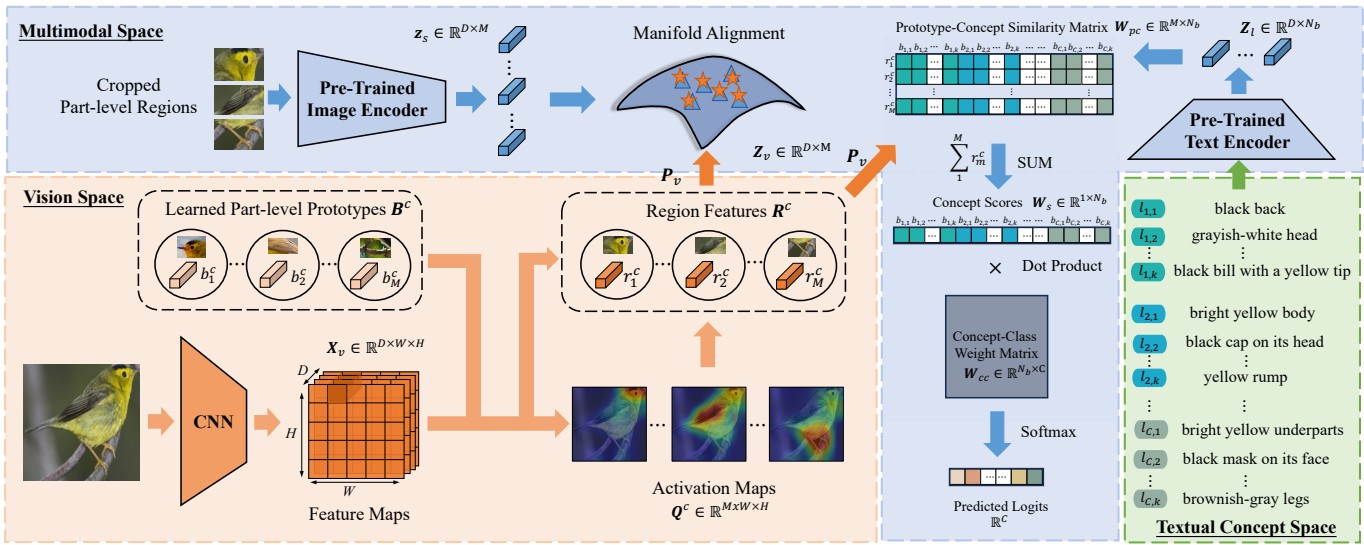

Figure 3: The overall architecture of our proposed Prototype-Concept Alignment Network (ProCoNet).

encoder $f_v(\cdot)$ denoted by $\mathbf{X}_v \in \mathbb{R}^{D \times W \times H}$. Here, $D$ represents the number of channels of the feature maps, while $H$ and $W$ denote the heights and widths of the feature maps. The vision encoder $f_v(\cdot)$ with paramters $\omega_{f_v}$ is composed of a convolutional network backbone (e.g. VGG, ResNet, and DenseNet) and additional 1×1 convolutional layers. Using extra 1×1 convolutional layers, the output of vision can be projected to the feature space of the same dimensionality as the multimodal space. In the ProtoPNet, the objective is to learn $M$ prototypes $\mathbf{B}^c = \{b_m^c\}_{m=1}^M \in \mathbb{R}^{D \times M}$ with $b_m^c \in \mathbb{R}^D$ for every category $c$, i.e. class-specific prototype. Each prototype unit $S_{b_m}$ first computes the squared $L_2$ distance $||\phi(\mathbf{X}_v) - b_m||_2^2$ between the prototypes $b_m$ and the all 1×1 patches of $\mathbf{X}_v$ and then convert to similarity by the function $\log((||\phi(\mathbf{X}_v) - b_m||_2^2 + 1)/(||\phi(\mathbf{X}_v) - b_m||_2^2 + \epsilon))$, where $\phi_{w,h}(\mathbf{X}_v)$ denotes the operation of taking the $(w, h)$-th location in the visual feature map. If the patch $\phi(\mathbf{X}_v)$ is closer to the prototype $b_m$, the function will produce a higher similarity score. Given an input image, an activation map $\mathbf{Q}_m^c$ with shape $W \times H$ of the $m$-prototype in class $c$ can be calculated by computing the similarity between each prototype and the features. The activation map can be up-sampled to the input image to illustrate how strongly the prototype $b_m^c$ is activated in the image and the score of the prototype can be obtained by performing a global max pooling on the similarity map. Finally, the output logits can be generated by multiplying the similarity score by the weights of the fully connected layer. Then, a softmax function is applied to calculate the probability that the current image belongs to all categories, with the highest probability corresponding to the category $\bar{y}$ predicted by the ProtoPNet network. For a pre-trained ProtoPNet, an image can re-represent its high-level features on different prototypes, while displaying prototype activation regions on the original image. Furthermore, it can make classification decisions based on the similarity with different prototypes.

For the predicted class $\bar{y}$ in the trained prototype network, we can encode each region with high activation for each prototype on

the feature maps based on the activation maps $\mathbf{Q}^{\bar{y}} = \{\mathbf{Q}_m^{\bar{y}}\}_{m=1}^M$ and prototypes $\mathbf{B}^{\bar{y}}$. The region features are achieved through non-linear feature encoding [18, 21, 23] as follows:

$$r_m' = \frac{1}{\sum_{w,h} q_{w,h}^m} \sum_{w,h} q_{w,h}^m (x_{w,h} - p_m^{\bar{y}}), r_m = \frac{r_m'}{||r_m'||_2}, \quad (1)$$

where $q_{w,h}^m = \phi_{w,h}(\mathbf{Q}_m^{\bar{y}})$ represents the activation value of each coordinate on the activation map and $x_{w,h} = \phi_{w,h}(\mathbf{X}_v)$ denotes the $(w, h)$-th location in the feature maps. Therefore, we obtained the region features $\mathbf{R}_v = \{r_m\}_{m=1}^M \in \mathbb{R}^{D \times M}$ of pixels assigned to the prototypes, which can represent the features of the corresponding prototype for predicted class $\bar{y}$ on the image.

**Region Feature Projection by Manifold Alignment.** To better project the region features of the visual space to the multimodal space for aligning with future and textual concepts, we first offline crop and resize the regions corresponding to the prototypes in trained ProtoPNet on the image. Then, we input them into the pre-trained vision language model CLIP's image encoder $f_I(\cdot)$ to obtain representations of the corresponding regions in the multimodal space. We denote the clip's image embedding of cropped part-level region as $\mathbf{Z}_s = \{z_m^s\}_{m=1}^M \in \mathbb{R}^{D \times M}$ with $z_m^s = \text{crop}(\mathcal{I}_v, r_m)$. Therefore, the region features $\mathbf{R}_v$ from the vision space and the cropped region features $\mathbf{Z}_s$ encoded by CLIP from the multimodal space respectively represent the features of the same image region in different spaces. Without any processing, the two sets of features would be different but semantically highly correlated manifold distributions. Our goal is to learn a projection $\mathbf{P}_v \in \mathbb{R}^{D \times D}$ to transfer the vision region features to a common subspace with the cropped region features encoded by CLIP in multimodal space such that the two distributions can be aligned semantically. The transformed features of vision region features $\mathbf{R}_v$ is denoted as $\mathbf{Z}_v = \mathbf{P}_v^\top \mathbf{R}_v$ in the multimodal space, which can serve as the representation of visual prototypes in the multimodal space. The specific method of learning this projection matrix $\mathbf{P}_v$ will be detailed in Section 3.3.

**Textual Conceptual Space Construction.** To construct a textual concept space recognizable by visual prototypes, our model designs a prompt generation method based on in-context learning to prompt the large language model, GPT-3 [3], to generate a set of $K$ textual concepts describing visual information for each category. In our model, we collected a total of $N_b$ textual concepts $l_{c,k}$ from the GPT-3. These textual concepts are encoded by the text encoder $f_L(\cdot)$ of CLIP, ultimately forming a concept representation $Z_l \in \mathbb{R}^{D \times N_b}$. The specific method for generating prompts will be described in Section 3.2.

**Visual Prototypes and Textual Concepts Alignment.** Given the projection $\mathbf{Z}_v$ of visual region features in multimodal space and the textual concept representation $\mathbf{Z}_l$, we first calculate the cosine similarity between both to construct a prototype-concept similarity matrix $\mathbf{W}_{pc} \in \mathbb{R}^{M \times N_b}$. The value at each position $(m_1, n_b)$ in the matrix is calculated by $\mathbf{W}_{pc}^{(m_1, n_b)} = \mathbf{z}_m^{v\top} \mathbf{z}_{n_b}^l / (||\mathbf{z}_m^v||_2 \cdot ||\mathbf{z}_{n_b}^l||_2)$ and each element represents the score of different visual prototypes of the image on textual concepts. Then we sum over the dimensions of the prototypes to obtain a score vector $W_s \in \mathbb{R}^{1 \times N_b}$, reflecting the total score of all prototypes in each concept. To establish a mapping from concept scores to final predictions, we learn a linear function for predicting categories, where the weights of the linear layer are denoted as $\omega_{cc} \in \mathbb{R}^{N_b \times C}$, encoding the correlation between concepts and categories. Furthermore, due to the inherent category prior information in textual concepts, we initialize the concept-category matrix $\omega_{cc}$ using linguistic priors. If the textual concept $l_{c,k}$ is within category $c$, we initialize the elements in the weight matrix related to category $c$ and concept $l_{c,k}$ as 1, otherwise as -0.5. Finally, the prediction based on concepts can be formalized by $\hat{y} = \arg\max(\sigma(W_s \cdot \omega_{cc}))$, where $\sigma(\cdot)$ is the softmax activation function.

**Main Objective Function.** The main objective of our model is to learn visual prototypes that can align with textual concepts. To learn part-level visual prototypes, the patches of feature maps in the vision space are clustered on at least one semantically similar prototypes of the ground truth category and separated from prototypes of the other category. Therefore, the introduced clustering and separation losses are formalized in [4] as:

$$\mathcal{L}_{clst} = \frac{1}{N} \sum_{i=1}^{N} \min_{j:b_j \in \mathbf{B}^{(y^i)}} \min_{x_v \in \phi(\mathbf{X}_v)} ||x_v - b_j||_2^2$$

$$\mathcal{L}_{sep} = -\frac{1}{N} \sum_{i=1}^{N} \min_{j:b_j \notin \mathbf{B}^{(y^i)}} \min_{x_v \in \phi(\mathbf{X}_v)} ||x_v - b_j||_2^2 \tag{2}$$

where $N$ is the number of training set and $y^i$ is the label of instance $i$. The optimization problem combined with the cross-entropy loss (CrsEnt) to penalize the misclassification of $\hat{y}$ can be formed as follows:

$$\mathcal{L}_{\text{total}} = \mathcal{L}_{\text{CrsEnt}} + \lambda_1 \mathcal{L}_{clst} + \lambda_2 \mathcal{L}_{sep} \tag{3}$$

where $\lambda_1$ and $\lambda_2$ is the hyper-parameter. A semantically meaningful concept space will be constructed under the constraints of these terms. Through the above optimization, visual prototypes can be discovered in the vision space.

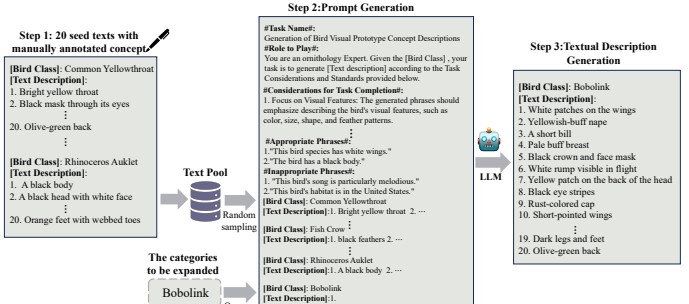

**Figure 4: The textual concept space generated by the large language model (LLM). We have designed a prompt generation method that combines in-context learning, and the generated textual concepts can be recognized by visual prototypes.**

## 3.2 Concept Space Construction

Previous work Labo [52] utilized simple prompts, such as *"describe what the [CLASS NAME] looks like"*, to prompt GPT-3 to generate related sentences, then split the sentences into shorter concepts using a T-5 model [35] fine-tuned on a small set of annotated sentence-concept pairs, and finally used a submodular optimization to filter out concept sets. LaBo pioneered the use of large language models to collect concept descriptions. However, the method involves cumbersome steps, requiring post-processing of sentences generated by large models with multiple models. Furthermore, even after multiple rounds of preprocessing, the concept set filtered by LaBo still contains many concepts that are mixed and cannot be visually recognized, such as *"This bird's song is particularly melodious"*, *"This bird's habitat is in the United States"*, *"Longevity, with some individuals living over 50 years"* and etc.

To address the aforementioned issues, we have designed a prompt generation strategy that incorporates in-context learning to prompt large language models, which enables the direct generation of textual concepts by the LLM without any post-processing, as illustrated in Figure 4. Specifically, we first annotate 20 text concepts for each category as seed texts for in-context learning, and then randomly select 8-12 from the text pool as guiding in context information during prompt generation. Next, we designed instructions specifically for prompting large language models to generate textual descriptions for visual prototypes. The instructions include the task name, the role the large language model needs to play, considerations for completing the task, task criteria, as well as examples of suitable and unsuitable phrases. Due to space limitations, detailed instructions will be presented in the appendix. The concatenated prompt is composed of instructions designed by us, in-context instances randomly sampled from the text pool, and the category names that need to be expanded. This merged prompt is inputted into the large language model, ultimately resulting in 20 text descriptions focusing on describing potential visual prototypes for the given category.

## 3.3 Manifold Alignment

The objective of the classical manifold alignment algorithm is to learn a projection that maps features from different spaces to a

common subspace simultaneously matching the corresponding instances and preserving the local geometry of each manifold. Matching the corresponding instance implies that instances with comparable semantic meanings from distinct manifolds will be projected onto neighboring regions following transformation, while preserving the local geometry implies that the relationship of the neighborhood of the instances in each manifold will remain consistent after transformation. Therefore, solving the alignment problem without correspondence needs to find two unknown variables, i.e.the correspondence, and the projection matrix. In our work, we aim to learn a projection matrix through manifold alignment to better map the vision space to the multimodal space while preserving the geometric relationships among the original region features in vision space.

**Constructing the Correspondence.** In our model, there is a clear one-to-one correspondence between the region feature $r_m$ of the visual space and the cropped regions $z_m^s$ encoded by CLIP corresponding to the visual prototypes. Therefore, we can leverage this property to construct the corresponding matrix as follows:

$$W_{vs}^{(m_1,m_2)} = \begin{cases} 1 & \text{if} \quad m_1 = m_2, \\ 0 & \text{otherwise} \end{cases} \tag{4}$$

where $W_{vs}^{(m_1,m_2)} \in \mathbb{R}^{M \times M}$ is the element of the correspondence matrix $\mathbf{W}_{vs}$. If $m_1$-th visual region feature is corresponding with $m_2$-th cropped regions encoded by CLIP, $W_{vs}^{(m_1,m_2)}$ is set to 1 or vice versa.

**Objective Function of Manifold Alignment.** In our work, the manifold alignment is used to make visual region features and cropped CLIP-encoded region features with similar semantics close after projected to the common subspace. To achieve the above goals, the objective function can be defined as follows:

$$\min_{\mathbf{P}_v} J(\mathbf{P}_v) = \frac{1}{M} \sum_{m_1=1}^{M} \sum_{m_2=1}^{M} W_{vs}^{(m_1,m_2)} ||z_{m_1}^v - z_{m_2}^s||_2^2, \tag{5}$$

where $M$ is the number of corresponding pairs, i.e. the number of element $W_{vl}^{(i,j)}$ equals 1 in correspondence matrix $\mathbf{W}_{vl}$. $z_{m_1}^v$ presents the projection of $r_{m_1}$ in multimodal space . If $r_{m_1}$ and $z_{m_2}^s$ correspond to each other in original space, the distance between $z_{m_1}^v$ and $z_{m_2}^s$ will be penalized in projected space.

In the common subspace unrolled by the manifolds, the nearest neighbor relationships of each manifold are expected to be protected, i.e. the local geometry should not be destroyed. In many manifold learning algorithms [6, 47], the local geometry is characterized by computing the local adjacency weight matrix, i.e. the similarity between instances in the manifold. According to the previous work [1, 41], the image and sentence structure are encoded by the similarities of the visual features with different locations and language features with different words. In our work, for preserving the local geometry, the orthogonal constraints are added on the projection matrix $\mathbf{P}_v$, i.e. $\mathbf{P}_v^\top \mathbf{P}_v = I$. Under the constraint of projection matrix orthogonality, the local geometry can be preserved and the proof is as follows:

$$d(\mathbf{z}_{m_1}^v, \mathbf{Z}_{m_2}^v) = ||\mathbf{P}_v^\top \mathbf{r}_{m_1} - \mathbf{P}_v^\top \mathbf{r}_{m_2}||_2^2 = \mathbf{P}_v^\top \mathbf{P}_v ||\mathbf{r}_{m_1} - \mathbf{r}_{m_2}||_2^2$$
$$= ||\mathbf{r}_{m_1} - \mathbf{r}_{m_2}||_2^2. \tag{6}$$

The original structure of visual features will be preserved with the orthogonal constraint in the projected common subspace.

For matching the semantically similar features among manifolds and protecting the local geometry of each manifold, we can combine the objective function Equation 5 with the orthogonal constraint. Firstly, we convert the Equation 5 to the following form:

$$\min_{\mathbf{P}_v} J(\mathbf{P}_v) = \text{tr}(\mathbf{P}_v^\top \mathbf{R}_v \mathbf{D}_v \mathbf{R}_v^\top \mathbf{P}_v + \mathbf{Z}_s \mathbf{D}_s \mathbf{Z}_s^\top - 2\mathbf{P}_v^\top \mathbf{R}_v \mathbf{U}_{vs} \mathbf{Z}_s^\top) \tag{7}$$

where $\mathbf{U}_{vs} = \mathbf{W}_{vs}/M$, $\mathbf{D}_v \in \mathbb{R}^{M \times M}$ is a diagonal matrix with the element $\mathbf{D}_v(m_1, m_1) = \sum_{m_2=1}^{M} \mathbf{U}_{vs}(m_1, m_2)$. The similar with $\mathbf{D}_s \in \mathbb{R}^{M \times M}$ is a diagonal matrix with the element $\mathbf{D}_s(m_2, m_2) = \sum_{m_1=1}^{M} \mathbf{U}_{vs}(m_1, m_2)$ and $\text{tr}(\cdot)$ is the trace.

However, considering the orthogonal constraint of projection matrices, the first and second terms are irrelevant to $\mathbf{P}_v$. The final manifold alignment problem can be transformed into solving the following optimization problem with constraints:

$$\min_{\mathbf{P}_v} J(\mathbf{P}_v) = -\text{tr}(2\mathbf{P}_v^\top \mathbf{Z}_v \mathbf{U}_{vs} \mathbf{Z}_s^\top)$$
$$\text{s.t.} \quad \mathbf{P}_v^\top \mathbf{P}_v = I \tag{8}$$

Solving the above optimal solution is difficult directly since the orthogonal constraints can lead to many local optimal solutions. The traditional approach utilizes Singular Value Decomposition (SVD) to solve the above equations, such as Canonical Correlation Analysis (CCA) [14]. However, this method relies on local batch data, requiring corresponding cropped region features from the multimodal space for each projection computation. Therefore, we propose a new solution to the aforementioned orthogonal problem, i.e. Equation (8), which can search for the global optimal solution across all data and does not require CLIP-encoded regional features during test inference. Specifically, we solve the optimization problem based on the Cayley transform and optimize the projection matrices on the Stiefel manifold in our work. In the Stiefel manifold, the feasible set $\mathcal{P} = \{\mathbf{P} \in \mathbb{R}^{C \times C} : \mathbf{P}^\top \mathbf{P} = I\}$ and projection matrix $\mathbf{P}_v$ is updated by a curvilinear search [51] in the feasible set through the Cayley transform:

$$\mathbf{P}_v^{(t+1)} = (I + \frac{\eta_v}{2} \mathbf{A}_v)^{-1} (I - \frac{\eta_v}{2} \mathbf{A}_v) \mathbf{P}_v^{(t)} \tag{9}$$

where $t$ is the present step, $t + 1$ is the next step, $\mathbf{A} = \mathbf{G}(\mathbf{P}^{(t)}) - \mathbf{P}^{(t)} \mathbf{G}^\top$ is a skew-symmetric matrix obtained by Cayley transform, $\mathbf{G}$ is the gradient of the objective function and $\eta$ is the learning rate. We can derive the closed-form solution for the gradient of the objective function as follows:

$$\mathbf{G} = -2\mathbf{Z}_v \mathbf{U}_{vs} \mathbf{Z}_s^\top \tag{10}$$

The stochastic gradient of a mini-batch is calculated to replace $\mathbf{G}$ in each step and momentum is applied to accelerate and stabilize the stochastic gradient. Through manifold alignment, we can obtain a projection matrix to project region features from the visual space onto the cropped region features in the multimodal space derived from the same visual prototype. These features share similar semantics, and the projection results will tightly lie on the aligned manifold in the multimodal space.

## 3.4 Implementation

Our model implementation consists of three stages. In the first stage, we pretrain a ProtoPNet and encode the images cropped from prototypes using the CLIP image encoder. In the second stage, we use a large language model to construct a set of textual concepts and encode them using the CLIP text encoder. In the third stage, we train our ProCoNet model to optimize the two optimization problems described in the previous sections. The first optimization problem is to learn the projection matrices $\mathbf{P}_v$ to align cross-modal manifold and the second optimization problem is the main objective to learn visual prototypes. We optimize the framework by alternating optimization that two objectives are optimized in turns. Algorithm 1 provides details of alternating optimization. $N_{epoch}$ denotes the number of training epochs; $\eta_f$, $\eta_{add}$, $\eta_B$ denotes the learning rate.

---

**Algorithm 1:** Alternating Optimization Algorithm

---

1 **Input:** $\mathcal{D} = \{\mathcal{I}_v^i, y^i\}_{i=1}^N, \{\mathcal{I}_l^j\}_{j=1}^{N_b}$
2 **Optimization Variables:** $\omega_{f_v}$, $\omega_{cc}$, $\mathbf{P}_v$, $\mathbf{B}$
3 **Parameters**: $\beta$, $\eta$
4 **for** $n_{epoch} = 1$ **to** $N_{epoch}$ **do**
5     **for** $t = 1$ **to** $T$ **do**
6         Sample a mini-batch $\{\mathcal{I}_v^i, y^i\}_{i=1}^m$ from $\mathcal{D}$
7         $\mathbf{B} \leftarrow \mathbf{B} - \eta_B \nabla_B \mathcal{L}_{\text{total}}(\mathcal{I}_v^i, y^i)$
8         $\omega_{f_v} \leftarrow \omega_{f_v} - \eta_{f_v} \nabla_{\omega_{f_v}} \mathcal{L}_{\text{total}}(\mathcal{I}_v^i, y^i)$
9         Calculate the gradients of $\mathbf{G}$ by Eq. 10
10         Accumulate the gradients by exponential moving
           average $\mathbf{G}' \leftarrow \beta\mathbf{G}' + (1 - \beta)\mathbf{G}$
11         **if** $t \bmod 30 = 0$ **then**
12             Calculate learning rates $\eta_v$ by curvilinear search
               refering to [51]
13             Update $\mathbf{P}_v$ by Eq. 9

---

## 4 Experiments

In the experiments, two case studies are conducted to evaluate our modal with other interpretable models in terms of accuracy and interpretability. The first case study is the bird species identification with 200 bird species on the CUB-200-2011 datasets [46], which is popular on the prototype-based concept learning [4, 30, 50]. The second case study is the flower species identification with 102 flower species on the Oxford Flowers datasets [31]. We adopt different CNN architectures as encoders in the vision space and verify the generality of our model for different visual encoders.

**Evaluation Metric**: We evaluate our model in terms of accuracy and interpretability for fine-grained image recognition. For accuracy, we compare the predicted label with the ground truth category to calculate the top-1 accuracy as considered in the previous interpretable image recognition tasks. However, interpretability has not been quantified in the previous ProtoNet and the extension methods. Inspired by the previous part discovery for fine-grained recognition [18], we designed different quantitative metrics of interpretability schemes for datasets with different annotations. In the CUB-200-2011 dataset, there are 15 part landmarks for each image and we measure the object part localization error by comparing the response region of the learned semantic concept with

**Table 1: Top: Results of accuracy comparison on cropped bird images of CUB-200-2011 with different vision encoders; Bottom: Results of landmark localization on cropped bird images of CUB-200-2011. Normalized L2 distance (%) is reported.**

| Method | VGG16 | VGG19 | ResNet34 | ResNet152 | Dense121 | Dense161 |
|---|---|---|---|---|---|---|
| Baseline-CNN | 73.3% | 74.6% | **81.2%** | **81.8%** | 80.2% | 81.2% |
| ProtoPNet [4] | 76.9% | 78.2% | 78.4% | 78.5% | 79.1% | 80.5% |
| **Ours** | **78.3%** | **79.3%** | 80.8% | 79.9% | **80.9%** | **81.8%** |
| Method | VGG16 | VGG19 | ResNet34 | ResNet152 | Dense121 | Dense161 |
| ProtoPNet [4] | 19.7 | 19.3 | 16.4 | 16.3 | 16.2 | 16.4 |
| **Ours** | **18.7** | **18.7** | **15.0** | **14.8** | **15.3** | **16.2** |

the annotated part landmarks. The part localization error has been adopted by [18, 19]. For the datasets without part annotations such as Flower-102, we adopt the protocol of Pointing Game [55], which is a popular method to quantify interpretability in visualization methods [40, 55], and calculate the object localization error using the annotated segmentation. The detailed metrics are described in the supplement.

### 4.1 Case study1: bird species identification

**Dataset.** Caltech-UCSD Birds-200-2011 [46] (CUB-200-2011) is a dataset of 200 bird species for bird species recognition and contains 5,994/5,794 images for training/testing from 200 different bird species. Each image contains a species label, 15 bird landmarks, and the bounding box of the bird.

**Recognition Results (Accuracy).** We present our results of recognition accuracy with different CNN vision encoders on cropped bird images at the top of Table 1, comparing them with CNN baselines and ProtoPNet. Additionally, we compared our results with other single-modality ProtoPNet-based models in the appendix. Across various CNN vision encoders, ProCoNet achieves higher classification accuracy than ProtoPNet, which learns visual prototypes through single-modality learning. Guided by textual concepts, visual prototype learning can improve accuracy by up to 2.4%.

**Localization Results (Interpretability).** The part localization error is evaluated on various CNN vision encoders and we compare the results to ProtoPNet at the bottom of Table 1. Our model greatly reduces the localization error of ProtoPNet (0.93% on average). The results provide quantifiable evidence of the interpretability of our model.

**Reasoning Process.** Figure 5 illustrates the reasoning process of our model. Through concept visualization, we observe that our model learns the typical features of the Green Jay, such as the distinctive blue head, green wings, and black throat, solely from label supervision. For a given test image, activation maps are generated by computing the distances between various regions of the image and the prototypes. Additionally, similarity maps are obtained by calculating the similarity scores between regions and textual concepts. Finally, these similarity scores are weighted by concept weights and contribute to the final classification. Compared to the traditional ProtoPNet reasoning process, our model not only highlights regions with high prototype activations but also understands the corresponding textual concepts of visual prototypes, thus providing better interpretability.

## Why is the bird classified as a Green Jay?

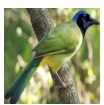

Evidence for this bird being a Green Jay:

| Original image | Prototype | Training image where concept comes from | Activation map | Textual Concepts (display top 3) | Similarity Score | | Class connection | | Contribution to logits |
|---|---|---|---|---|---|---|---|---|---|
| | | | | blue head | 0.2853 | × | 0.8587 | = | 0.2450 |
| | | | | black mask around the eyes | 0.2769 | × | 0.8280 | = | 0.2293 |
| | | | | blue feathers | 0.2598 | × | 0.8387 | = | 0.2179 |
| | | | | green wings withe primaries | 0.2859 | × | 0.8612 | = | 0.2462 |
| | | | | green and yellow feathers | 0.2777 | × | 0.8357 | = | 0.2320 |
| | | | | bright yellow underparts ... | 0.2583 | × | 0.7921 | = | 0.2045 |
| | | | | black markings on its face and throat | 0.2871 | × | 0.8268 | = | 0.2373 |
| | | | | black feet and legs | 0.2790 | × | 0.7479 | = | 0.2087 |
| | | | | black bill ... | 0.2608 | × | 0.7770 | = | 0.2026 |

Total points to the Green Jay: 15.317

Figure 5: The interpretable reasoning process to identify the species of a bird.

**Ablation Study.** We conducted an ablation study on the CUB dataset to evaluate the components of the manifold alignment. We designed two variants: one without aligning the region images and another using L1 loss to optimize the projection matrix without enforcing orthogonality constraints. We compared these variants using VGG-19 as the visual encoder. As shown in Table 2, omitting the alignment of cropped region images leads to a decrease in model accuracy and interpretability. Additionally, compared to optimizing the projection matrix with L1 loss, manifold alignment optimization of the projection matrix improves both model accuracy(+0.8%) and interpretability(+0.4%).

Table 2: Ablation study about Manifold Alignment.

| Methods | Acc(%) | Err(%) |
|---|---|---|
| w/o align | 76.6 | 20.7 |
| L1 | 78.4 | 19.1 |
| Ours | 79.3 | **18.7** |

## 4.2 Case study2: Flower species identification

**Dataset.** The Oxford Flowers-102 [31] is a dataset of 102 different categories for flower species identification. There are 10 images per class in the training and validation set (totaling 1020 images each), and the remaining 6149 images (minimum 20 per class) in the test set. Each image contains a species label and a segmentation of the flower.

**Recognition Results (Accuracy).** Our results on recognition are summarized at the top of Table 3. Since flowers do not have obvious object parts, they are more judged based on shapes and colors, thus, the ProtoPNet-based method, which is good at learning object parts, is not as good as the baseline model. Our model can improve accuracy by 0.6%-2.3% for ProtoPNet.

**Localization Results (Interpretability).** Further, the Pointing game results are reported at the bottom of Table 3. Our model achieves a lower localization error compared to ProtoPNet. To a certain extent, our models can capture more foreground concepts.

## Why is the flower classified as a Pink Primrose?

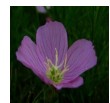

Evidence for this flower being a Pink Primrose:

| Original image | Concept | Training image where concept comes from | Activation map | Textual Concepts (display top 3) | Similarity Score | | Class connection | | Contribution to logits |
|---|---|---|---|---|---|---|---|---|---|
| | | | | a yellow center | 0.2840 | × | 0.7962 | = | 0.2261 |
| | | | | light gradient toward the center | 0.2738 | × | 0.7167 | = | 0.1962 |
| | | | | delicate and airy appearance | 0.2549 | × | 0.7022 | = | 0.1790 |
| | | | | light pink petals | 0.2870 | × | 0.7389 | = | 0.2120 |
| | | | | frilly edges | 0.2794 | × | 0.7416 | = | 0.2072 |
| | | | | very feminine shade of pink ... | 0.2738 | × | 0.8253 | = | 0.2260 |
| | | | | a star shape | 0.2840 | × | 0.7121 | = | 0.2022 |
| | | | | small yellow center surrounded by pink petals | 0.2826 | × | 0.7389 | = | 0.2088 |
| | | | | a yellow center ... | 0.2762 | × | 0.7962 | = | 0.2200 |

Total points to the Pink Primrose: 12.317

Figure 6: The interpretable reasoning process to identify the species of a flower.

Table 3: Top: Results of accuracy comparison on flower images of the Oxford Flowers-102 with different vision encoders; Bottom: Results of Pointing Game on flower images of the Oxford Flowers-102. Hit rate (%) is reported.

| Method | VGG16 | VGG19 | ResNet34 | ResNet152 | Dense121 | Dense161 |
|---|---|---|---|---|---|---|
| Baseline-CNN | **89.3%** | **89.7%** | **93.6%** | **94.3%** | **94.9%** | **95.1%** |
| ProtoPNet [4] | 87.0% | 87.7% | 91.6% | 90.6% | 92.0% | 92.7% |
| **Ours** | 88.2% | 88.3% | 92.7% | 92.9% | 92.6% | 93.5% |
| Method | VGG16 | VGG19 | ResNet34 | ResNet152 | Dense121 | Dense161 |
| ProtoPNet [4] | 80.0 | 80.1 | 87.3 | 85.5 | 81.2 | 80.2 |
| **Ours** | **82.0** | **82.3** | **88.4** | **86.1** | **81.6** | **80.6** |

**Reasoning Process.** Figure 6 shows the reasoning process to identify a flower. The model can accurately learn bright purple petals and white pistil in the center. Since flowers lack sufficient object parts and there are few images in the training set, there is some redundancy in concepts.

## 5 Conclusion

In this paper, we introduce ProCoNet, a novel framework for interpretable image recognition that uniquely integrates semantic concepts from both visual and linguistic modalities. This first-of-its-kind system combines introspective and justification explainability, offering a new form of interpretability. Through our innovative prompt generation strategy, we have simplified the process of extracting text concepts and enhanced the clarity of descriptions. Additionally, we have introduced a manifold alignment method optimized via the Cayley transform on the Stiefel manifold, which, together with an alternating optimization algorithm, ensures robust alignment and concept learning. Extensive testing with interpretability metrics has demonstrated that incorporating text concept guidance significantly enhances the model's accuracy and interpretability, setting a new standard for interpretable AI systems in image recognition.

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
