# OpenReview forum: "Align2Concept: Language Guided Interpretable Image Recognition by Visual Prototype and Textual Concept Alignment"
_acmmm.org/ACMMM/2024/Conference — MM2024 Poster_

### Official Review · Reviewer_rEwH · 2024-05-24

**Rating:** 3
**Confidence:** 2

**Summary:**

The paper "Align2Concept: Language Guided Interpretable Image Recognition by Visual Prototype and Textual Concept Alignment" introduces a novel approach to interpretable image recognition by proposing the Prototype-Concept Alignment Network (ProCoNet). This model uniquely integrates visual prototypes and textual concepts using the pre-trained CLIP model to align visual features from images with corresponding textual descriptions. The framework aims to enhance both the interpretability and accuracy of image recognition models by leveraging linguistic information alongside visual data.

**Strengths:**

The primary strengths of this paper lie in its innovative approach and robust technical methodology. The introduction of ProCoNet represents a significant advancement in the field of interpretable AI, combining insights from cognitive science with cutting-edge vision-language models. The use of manifold alignment to project visual features into a multimodal space, aligning them with textual concepts, is a technically sound and well-justified approach. Furthermore, the optimization of the projection matrix using the Cayley transform on the Stiefel manifold is a noteworthy technical contribution. The paper is also commendable for its comprehensive evaluation, using the CUB-200-2011 and Oxford Flower datasets to demonstrate that ProCoNet achieves higher accuracy and better interpretability compared to existing single-modality models. The clarity of the presentation, with well-structured explanations and illustrative diagrams, aids in understanding the proposed framework. Additionally, the potential applications of this method in high-stakes domains, such as medical diagnosis and autonomous driving, where interpretability is crucial, underscore its practical relevance.

**Limitations:**

However, the paper does have several limitations. One major concern is the complexity of the proposed framework, which involves multiple stages of training and optimization. This complexity might hinder the reproducibility and practical implementation of the model. The effectiveness of ProCoNet is demonstrated on specific datasets, and it remains to be seen how well the model performs on more diverse datasets with different characteristics. The process of generating textual concepts relies heavily on large language models (LLMs), which may introduce biases inherent in these models, potentially affecting the performance and interpretability of the system. While the paper introduces new metrics for interpretability, it does not extensively compare them with existing metrics, which would have provided stronger validation of the proposed approach. Additionally, the scalability of the ProCoNet framework is not thoroughly discussed, raising concerns about its application to larger datasets or real-time scenarios. Finally, the paper focuses primarily on academic benchmarks without addressing real-world deployment challenges such as computational costs, inference time, and integration with existing systems.

**Suitability:**

2

---

### Official Review · Reviewer_TwDb · 2024-05-26

**Rating:** 6
**Confidence:** 3

**Summary:**

The paper introduces ProCoNet, Prototype-Concept Alignment Network, a novel method for enhancing interpretability and accuracy in image recognition by aligning visual prototypes with textual concepts. ProCoNet uses a visual encoder to extract regional image features and a large language model to generate relevant textual concepts. These are aligned using the pre-trained vision-language model CLIP through a manifold alignment method optimized with the Cayley transform on the Stiefel manifold.

Methodology involves decomposing images into visual prototypes, generating textual concepts using prompt strategies, and aligning these modalities in a common subspace. This alignment preserves the local geometry of the original features, ensuring semantic consistency.

Experiments on the CUB-200-2011 and Oxford Flowers-102 datasets show that ProCoNet achieves higher accuracy and better interpretability compared to single-modality models like ProtoPNet. The results demonstrate improved part localization and reduced localization errors.

**Strengths:**

The theoretical foundation of ProCoNet is robust, utilizing distribution matching for aligning visual and textual modalities. The optimization of the projection matrix using the Cayley transform on the Stiefel manifold is mathematically sound, ensuring that the local geometry of the feature space is preserved. This rigorous approach guarantees the technical correctness and reliability of the model.

The paper introduces a groundbreaking approach by integrating multi-modal information for image recognition. ProCoNet aligns visual prototypes with textual concepts, a novel method that enhances both interpretability and accuracy. This approach mimics human cognitive processes, making the model more intuitive and effective in understanding and explaining its decisions.

**Limitations:**

The related work section of the paper is currently kind of lengthy and lacks a focus on the most relevant and recent studies. This makes it difficult for readers to quickly grasp the essential background and context. Additionally, the literature review omits important recent works on multi-modal interpretability, which are crucial for the core contributions of this paper. Also, the connection between related works and the proposed method is not always explicit, which can obscure the significance of ProCoNet's innovations.

In the experimental section, several limitations need to be addressed to improve the clarity and comprehensiveness of the presented research. The paper lacks detailed explanations of key hyperparameters such as mini-batch size, training steps, and initial learning rate, which are essential for replicating the experiments. Additionally, the evaluation metrics are not clearly defined within the context of the study's objectives, making it challenging to understand their significance. The comparative analysis and specific process descriptions are also insufficiently detailed, limiting the understanding of how ProCoNet outperforms existing models.

**Suitability:**

3

---

### Official Review · Reviewer_aSKy · 2024-05-30

**Rating:** 4
**Confidence:** 4

**Summary:**

This paper proposes an approach, ProCoNet to learn semantic concepts based on Prototype theory via both linguistic and visual modalities, implementing for interpretable image recognition. Specifically, it follows a two-stream model in which a manifold alignment method is utilised among the language feature manifold and visual feature manifold by learning the projection matrices through Cayley transform on the Stiefel manifold. The author further verifies the validity of the model by testing on the CUB-200-2011 and Oxford Flowers datasets.

**Strengths:**

- The problem of concern in this paper has a practical background and is reasonably motivated. It has research value.
- The structure of the paper is clear, the logic is smooth, and the proposed method is clearly demonstrated.
- The idea of semantically meaningful concepts to be interpretable is promising.
- Given the empirical experiments, the proposed ProCoNet shows performance improvements over the baseline model in terms of both accuracy and interpretability.

**Limitations:**

- The analysis of manifold algorithms is somewhat weak, as learning semantic concepts is straightforward from both text and image features in the original space. Also, authors should provide hyperparameter details (optimiser, learning rate for both text and visual encoder, coefficients, etc.) for reproducibility.

- Also, how ProtoNet is different from a multi-manifold scenario? I would have liked to see some instances of images and concepts/texts that are close to each other.

- The authors should justify and provide explanations on how the value of manifold alignment, in particular, ProCoNet as compared to other approaches such as MCD [4] and Prototypical relevance propagation [5].

- The baseline comparisons [2,3] and experiments can be strengthened with some benchmarked datasets [1] which employ manifold learning, this could well reason with the shown model performance.

```
1. Zhao, L., Zheng, K., Zheng, Y., Zhao, D., & Zhou, J. (2023, July). RLEG: vision-language representation learning with diffusion-based embedding generation. In International Conference on Machine Learning

2. Mugnai, D., Pernici, F., Turchini, F., & Del Bimbo, A. (2022). Fine-grained adversarial semi-supervised learning. ACM Transactions on Multimedia Computing, Communications, and Applications (TOMM)

3. Korsch, D., Bodesheim, P., & Denzler, J. (2021, September). End-to-end learning of fisher vector encodings for part features in fine-grained recognition. In DAGM German Conference on Pattern Recognition

4. Vielhaben, J., Bluecher, S., & Strodthoff, N. (2023). Multi-dimensional concept discovery (MCD): A unifying framework with completeness guarantees. Transactions on Machine Learning Research.

5. Gautam, S., Höhne, M. M. C., Hansen, S., Jenssen, R., & Kampffmeyer, M. (2023). This looks more like that: Enhancing self-explaining models by prototypical relevance propagation. Pattern Recognition.
```

**Suitability:**

3

---

### Meta-Review · Area_Chair_fmQx · 2024-07-03

**Recommendation:** Accept (Poster)
**Confidence:** 4

**Metareview:**

This is a borderline submission with mixed reviews. There were some concerns related to the lack of clarity, which has been addressed well in the author's rebuttal. Reviewer rEwH's concerns remain, but the AC believes these concerns are too challenging to address in a single paper. Considering the idea is well-motivated and experimental results are promising, the final decision is a borderline acceptance.